# Research on an Improved Sliding Mode Observer for Speed Estimation in Permanent Magnet Synchronous Motor

**Zhiqiang Liu \* and Wenkai Chen**

College of Automotive and Mechanical Engineering, Changsha University of Science & Technology, Changsha 410114, China; cwk1022@163.com
\* Correspondence: lzq0228@126.com

**Abstract:** Aimed at the problems of system chattering and large observation errors in the sensorless control of a permanent magnet synchronous motor (PMSM) based on a traditional sliding mode observer (SMO), a combined reaching law algorithm based on the exponential reaching law and arcsine saturation function reaching law is proposed to improve the sliding mode observer. An appropriate positive real number is taken and compared with the product of controller gain and stator current observation error to judge the system position in sliding mode motion. In the early stage of sliding mode motion, the exponential reaching law is utilized, and then, in the latter and stable stages of sliding mode motion, the arcsine saturation function reaching law is used. The stability of the observer is proved by Lyapunov theory. The simulation and experimental data show that the speed error of the sliding mode observer based on the combined reaching law is reduced by 80% compared with the traditional sliding mode observer, and the chattering problem is also improved.

**Keywords:** permanent magnet synchronous motor (PMSM); sliding mode observer (SMO); inverse saturation function; composite reaching law





## 1. Introduction

Because of its high energy density, high efficiency, and wide speed range in the constant power zone, PMSMs are widely employed in electric vehicle drive systems [1–3]. The precise rotor position and speed data typically provided through sensors are required for high-performance PMSM control. Because of the poor working environment of the electric vehicle drive system, adding sensors not only raises the hardware cost but also reduces the reliability of the system. To overcome these problems, research on sensorless control has become a hot topic in driving motors [4–7].

The high-frequency injection method, Kalman filter method, model reference adaptive method, and sliding mode observer method are some of the sensorless control algorithms used by PMSMs [8–15]. The sliding mode observer strategy is often used in sensorless control because of its minimal requirement for system model accuracy, insensitivity to parameter changes and external disturbances, and good resilience [16,17]. The typical sliding mode observer uses the discontinuous sign function to provide sliding mode motion [18]; however, there is a mechanical delay in engineering applications, resulting in unavoidable buffeting. Many continuous switching functions have been devised in advanced research to replace the traditional discontinuous symbolic function to avoid chatterings, such as saturation [19], sigmoid function [20], and hyperbolic function [21]. The thickness of the boundary layer has two opposing impacts on the reaching law: the thicker the boundary layer, the less buffeting the system suffers over the sliding mode surface, but the lower the control accuracy, and vice versa [22].

According to the above analysis, this paper takes the interior PMSM vector control system with zero d-axis currents (id = 0) as the background to solve the chattering problem of the sliding mode system. An anti-sinusoidal saturation function with a variable boundary

layer is proposed to reduce the chattering of sliding mode system. Furthermore, to improve the convergence speed of the inverse sine saturation sliding mode function, an exponential function reaching law is proposed, which is combined with the inverse sine saturation function to form a combined reaching law for the sensorless vector control of PMSM. It not only suppresses the sliding mode chattering, but also speeds up the convergence speed of the system and improves the estimation accuracy of the motor speed.

## 2. Mathematical Model of PMSM

Taking the IPMSM as an example, in $\alpha\beta$ coordinates, the model is expressed as follows:

$$
\begin{aligned}
\frac{di_\alpha}{dt} &= -\frac{R}{L}i_\alpha + \frac{u_\alpha}{L} - \frac{E_\alpha}{L} \\
\frac{di_\beta}{dt} &= -\frac{R}{L}i_\beta + \frac{u_\beta}{L} - \frac{E_\beta}{L} \\
E_\alpha &= -\omega_r\psi_f \sin\theta_r \\
E_\beta &= \omega_r\psi_f \sin\theta_r
\end{aligned}
\tag{1}
$$

where $R$ is the stator resistance; $L$ is the stator inductance; $i_\alpha, i_\beta$ are the stators in the $\alpha$-axis and $\beta$-axis, respectively; $u_\alpha, u_\beta$ are the voltages in the $\alpha$-axis and $\beta$-axis, respectively; $E_\alpha, E_\beta$ are the stator back electro-motive forces in the $\alpha$-axis and $\beta$-axis, respectively; $\omega_r$ is the angular speed of the rotor; $\psi_f$ is the permanent magnet flux linkage; $\theta_r$ is the angular position of the rotor.

Equation (1) is abbreviated as:

$$
\frac{d}{dt}\begin{bmatrix} i_\alpha \\ i_\beta \end{bmatrix} = A\begin{bmatrix} i_\alpha \\ i_\beta \end{bmatrix} + B\begin{bmatrix} u_\alpha \\ u_\beta \end{bmatrix} + C\begin{bmatrix} E_\alpha \\ E_\beta \end{bmatrix}
\tag{2}
$$

where

$$
A = \begin{bmatrix} -\frac{R}{L} & 0 \\ 0 & -\frac{R}{L} \end{bmatrix} B = \begin{bmatrix} \frac{1}{L} & 0 \\ 0 & \frac{1}{L} \end{bmatrix} C = \begin{bmatrix} -\frac{1}{L} & 0 \\ 0 & -\frac{1}{L} \end{bmatrix}
\tag{3}
$$

## 3. Traditional Sliding Mode Observer

According to Equation (2) and sliding mode variable structure theory, the mathematical model of the sliding mode observer is as follows:

$$
\frac{d\hat{i}}{dt} = A\hat{i} + Bu + v
\tag{4}
$$

where '$\wedge$' represents the estimated value, and $v$ is the sliding mode control law.

$$
\hat{i} = \begin{bmatrix} \hat{i}_\alpha \\ \hat{i}_\beta \end{bmatrix} \quad v = -Ksign(\hat{i} - i)
\tag{5}
$$

where $K$ is a matrix of the observer gain; $sign$ is the switching function of the reaching law; and $K$ can be expressed as:

$$
K = \begin{bmatrix} \frac{k}{L} & 0 \\ 0 & \frac{k}{L} \end{bmatrix}
\tag{6}
$$

where $k$ is the switching gain of the observer.

Rewrite Equation (4) in the component form:

$$
\begin{aligned}
\frac{d\hat{i}_\alpha}{dt} &= -\frac{R}{L}\hat{i}_\alpha + \frac{u_\alpha}{L} - \frac{k}{L}sign(\hat{i}_\alpha) \\
\frac{d\hat{i}_\beta}{dt} &= -\frac{R}{L}\hat{i}_\beta + \frac{u_\beta}{L} - \frac{k}{L}sign(\hat{i}_\beta)
\end{aligned}
\tag{7}
$$

The stator current error in the $\alpha$-axis and $\beta$-axis can be obtained by subtracting Equation (7) from Equation (1), and it is as follows:

$$
\begin{aligned}
\frac{d\bar{i}_\alpha}{dt} &= -\frac{R}{L}\bar{i}_\alpha + \frac{E_\alpha}{L} - \frac{k}{L}sign(\bar{i}_\alpha) \\
\frac{d\bar{i}_\beta}{dt} &= -\frac{R}{L}\bar{i}_\beta + \frac{E_\beta}{L} - \frac{k}{L}sign(\bar{i}_\beta)
\end{aligned}
\tag{8}
$$

Let $\bar{i}_s$ be the error of the stator current between the observed and the actual value, i.e., $\bar{i}_s = \hat{i}_s - i_s$.

Take the sliding surface $S = \bar{i}_s$ and make the sliding surface $S = \bar{i}_s = x$, where $x$ is the state variable of the system, so $x$ is both the sliding surface and the state variable. For the convenience of analysis, $x$ is uniformly replaced in the following description. The traditional sliding mode reaching law adopts the exponential reaching law as follows:

$$
\dot{S} = -AS - Ksign(S)
\tag{9}
$$

where $\dot{S}$ is the derivative of $S$; $-AS$ is an exponential reaching term; $-Ksign(S)$ is a constant reaching term.

Because the system buffeting is mainly related to the constant velocity reaching term, a smaller matrix determinant $|K|$ value and a larger matrix determinant $|A|$ value are usually selected to reduce the influence of buffeting. $|A|$ is a constant, which depends on the parameters of the system. When the estimation errors of the state variable are large, the constant velocity reaching term makes the system move to the sliding mode surface at the same reaching velocity and maintain the sliding mode. At this time, the system still switches on the sliding mode surface at the constant velocity reaching rate and does not adaptively adjust the gain to change the reaching rate according to the change in the system state, resulting in large buffeting of the system on the sliding mode surface. Therefore, the system buffeting will affect the speed estimation, and the optimization effect of the traditional exponential reaching law is very limited.

When the system reaches the sliding mode surface under the action of the control law, the $S = \hat{i}_s - i_s$ approximates to 0, so the following formula can be deduced:

$$
\begin{cases}
E_\alpha = ksign(\bar{i}_\alpha) \\
E_\beta = ksign(\bar{i}_\beta)
\end{cases}
\tag{10}
$$

At this time, the back electromotive force (EMF) of the motor is determined by the sign function and the switching gain, as shown in Equation (10). Due to the discontinuity of the sign function, the back EMF contains a large number of high-frequency harmonics, which intensify the inherent buffeting in sliding mode control. Therefore, it is necessary to carry out low pass filtering on the estimated back EMF to obtain its smooth values, and then estimate the speed and position of the motor. The low pass filtering expression is as follows:

$$
\begin{bmatrix}
\hat{E}_\alpha \\
\hat{E}_\beta
\end{bmatrix}
=
\begin{bmatrix}
-E_\alpha + k \cdot sign(\bar{i}_\alpha))/\tau_0 \\
-E_\beta + k \cdot sign(\bar{i}_\beta))/\tau_0
\end{bmatrix}
\tag{11}
$$

where $\tau_0$ is the time constant of the low-pass filter.

After filtering, the estimated rotor position information can be extracted from the back EMF:

$$
\hat{\theta} = -\arctan\left(\frac{\hat{E}_\alpha}{\hat{E}_\beta}\right)
\tag{12}
$$

The back EMF will cause the phase delay after low pass filtering, which is closely related to the cutoff frequency of the filter. When the motor speed is high, there is obvious phase delay in the back EMF due to the large variety of current harmonics and flux density. On the contrary, back EMF does not have this situation. The lower the cutoff frequency is, the more serious the phase delay is. Therefore, when estimating the rotor position and speed of the motor, the rotor position needs to be compensated. The compensated rotor position is shown below:

$$\theta_e = \hat{\theta} + \arctan\left(\frac{\hat{\omega}}{\omega_c}\right) \tag{13}$$

where $\omega_c$ is the cutoff frequency of the low pass filter.

In order to obtain the speed information, the differential operation of Equation (12) is carried out:

$$\omega_e = \frac{d\theta_e}{dt} \tag{14}$$

where $\omega_e$ is the angular velocity of the rotor.

## 4. Combined Reaching Law and Sliding Mode Observer Design

Sliding mode control includes two processes: reaching motion and sliding mode. The sliding mode stage is robust to perturbation and external disturbance, but the reaching motion stage does not have this characteristic. In addition, there is a time lag in the process of changing from reaching motion to sliding mode, which causes high-frequency vibration or resonance of the system. Therefore, for the controlled object with unstable or uncertain parameters, with the traditional sliding mode variable structure control method, it is difficult to ensure its asymptotic stability.

### 4.1. Improved Sliding Mode Reaching Law

The mathematical expression of the constructed anti-sinusoidal saturation function is shown below:

$$sat(S) = \begin{cases} 1 & S \geq \frac{1}{\varepsilon} \\ \arcsin(\frac{\lambda S}{\varepsilon}) & |S| < \frac{1}{\varepsilon} \\ -1 & S \leq -\frac{1}{\varepsilon} \end{cases} \tag{15}$$

where $\lambda = \sin(1) \approx 0.8415$; $\varepsilon > 0$ is boundary layer thickness; $S = \bar{i}_s$ is sliding surface. The mathematical characteristic curve of the saturation function is shown in Figure 1.

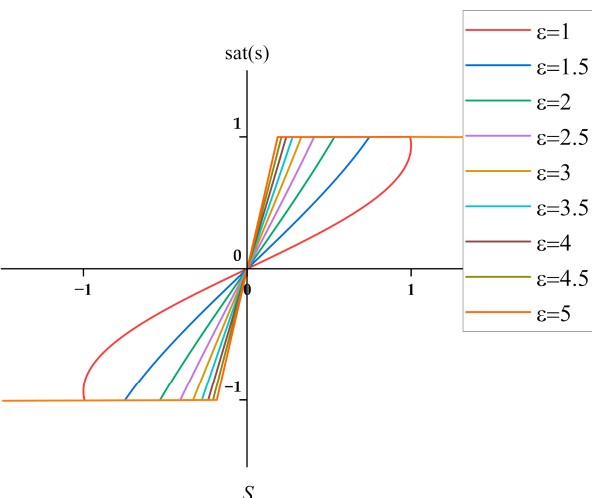

**Figure 1.** Inverse saturation function.

Because the stability of the system is related to the slope of the curve, the optimization effect of the system and the stability of the back EMF can be detected by modifying the boundary layer thickness $\varepsilon$. The smaller the boundary layer thickness is, the smoother the curve is, and the higher the control accuracy is. When the boundary layer thickness is different, the velocity of the system state variable towards the sliding mode surface is also different. The greater the thickness of the boundary layer is, the greater the buffering is. Its advantage is that $sat(S)$ has the characteristic of fast exponential function feedback in the boundary layer. After continuous adjustment, it is found that when $\varepsilon = 1.5$, the control law can quickly reach the sliding surface and reduce the chattering phenomenon of the system in the switching band; thereby the estimation accuracy of the motor speed and position information can be improved.

In order to further reduce the chattering caused by the constant velocity reaching term and improve the reaching speed of the state point outside the sliding mode, a variable gain reaching law is proposed based on Equation (15):

$$\overset{\bullet}{S} = -AS - K|S|\arcsin(\frac{\lambda S}{\varepsilon}) \tag{16}$$

where $k > 0$ is the switching gain. Through the new sliding mode reaching law, it can be seen that when the system state variable is far from the sliding surface ($S$ is large), it can approach the sliding surface at a large reaching speed. When the system state variable is close to the sliding surface, its reaching speed is gradually reduced due to the continuous decrease in $|S|$, which ensures a small gain to suppress the chattering phenomenon of the system entering the sliding surface. The phase trajectory of the new sliding mode reaching law is shown in Figure 2.

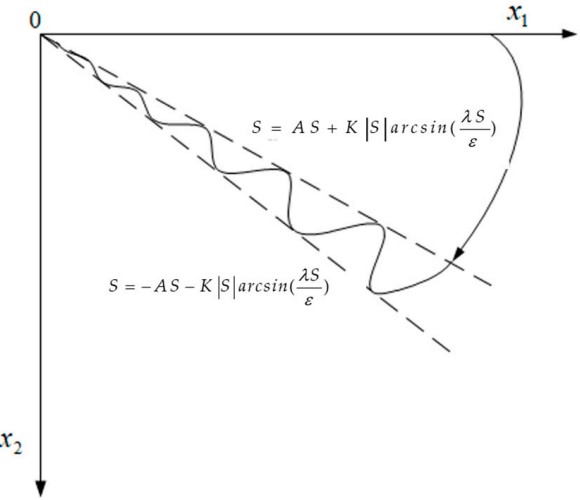

**Figure 2.** Improved sliding mode approach phase trajectory.

### 4.2. Design of Sliding Mode Observer with Combined Reaching Law

The switching band of the exponential reaching law is band shaped. In the steady state, the sliding mode function switches back and forth between its two sliding mode surfaces, thus resulting in large steady-state buffeting. When the new reaching law is used, the switching band consists of two rays passing through the origin and converging to the origin along the ray, which effectively reduces the chattering of the system in the steady state. However, for the new reaching law, when the system first enters the switching band, it produces larger chattering due to the larger value of $K|S|$.

If the exponential reaching law and the improved reaching law are combined, that is, in the early stage of sliding mode motion, the exponential reaching law is used, and in the later and steady-state stage of sliding mode motion, the improved reaching law can reduce the shortcomings of the two reaching laws and retain their advantages, so that the system

performance can reach its best. Select a positive real number $a_0$, when $K|S| > a_0$, using the exponential reaching law; when $K|S| \leq a_0$, the improved reaching law is used. When the value $a_0$ is large, the advantages of the improved reaching law will be weakened; when the value $a_0$ is small, the system may produce large chattering across the sliding surface.

By Equations (9) and (14), the combined reaching law of the system can be obtained:

$$\overset{\bullet}{S} = \begin{cases} -AS - K|S|\arcsin(\frac{\lambda S}{\varepsilon}) & K|S| > a_0 \\ -AS - Ksign(S) & K|S| \leq a_0 \end{cases} \tag{17}$$

According to Equations (7) and (17), the mathematical model of the new sliding mode observer is:

$$\begin{aligned} \frac{d\hat{i}_\alpha}{dt} &= -\frac{R}{L}\hat{i}_\alpha + \frac{E_\alpha}{L} + \frac{v_\alpha}{L} \\ \frac{d\hat{i}_\beta}{dt} &= -\frac{R}{L}\hat{i}_\beta + \frac{E_\beta}{L} + \frac{v_\beta}{L} \end{aligned} \tag{18}$$

In Equation (17):

$$\begin{cases} v_\alpha = Ksign(\hat{i}_\alpha) & \\ v_\beta = Ksign(\hat{i}_\beta) & K\left|\hat{i}_s\right| > a_0 \\ v_\alpha = K\left|\hat{i}_\alpha\right|\arcsin(\frac{\lambda}{\varepsilon}(\hat{i}_\alpha)) & \\ v_\beta = K\left|\hat{i}_\beta\right|\arcsin(\frac{\lambda}{\varepsilon}(\hat{i}_\beta)) & K\left|\hat{i}_s\right| \leq a_0 \end{cases}$$

By subtracting Equations (18) and (1), the state equation of the stator current error in a two-phase stationary coordinate system is obtained:

$$\begin{aligned} \frac{d\tilde{i}_\alpha}{dt} &= -\frac{R}{L}\tilde{i}_\alpha + \frac{E_\alpha}{L} + \frac{v_\alpha}{L} \\ \frac{d\tilde{i}_\beta}{dt} &= -\frac{R}{L}\tilde{i}_\beta + \frac{E_\beta}{L} + \frac{v_\beta}{L} \end{aligned} \tag{19}$$

The sliding mode switching surface $S(x) = \begin{pmatrix} \tilde{i}_\alpha & \tilde{i}_\beta \end{pmatrix}$ is designed based on the difference in current. When the state variables of the observer reach a steady state, there are $\frac{d\tilde{i}_\alpha}{dt} = 0$, $\frac{d\tilde{i}_\beta}{dt} = 0$. Based on the equivalent control principle of sliding mode control, the control quantity can be regarded as equivalent control quantity, and the extended back electromotive force can be obtained as follows:

$$\begin{cases} E_\alpha = Ksign(\tilde{i}_\alpha) & \\ E_\beta = Ksign(\tilde{i}_\beta) & K\left|\tilde{i}_s\right| > a_0 \\ E_\alpha = K\left|\tilde{i}_\alpha\right|\arcsin(\frac{\lambda}{\varepsilon}(\tilde{i}_\alpha)) & \\ E_\beta = K\left|\tilde{i}_\beta\right|\arcsin(\frac{\lambda}{\varepsilon}(\tilde{i}_\beta)) & K\left|\tilde{i}_s\right| > a_0 \end{cases} \tag{20}$$

To make the sliding mode control meet the requirements, it is necessary to select the appropriate $K$ value. According to Lyapunov stability theory, the Lyapunov function is selected as:

$$V = \frac{1}{2}\begin{pmatrix} \tilde{i}_\alpha & \tilde{i}_\beta \end{pmatrix}\begin{pmatrix} \tilde{i}_\alpha \\ \tilde{i}_\beta \end{pmatrix} \tag{21}$$

Only when $\overset{\bullet}{V} \leq 0$ can the system satisfy the stability condition. A derivation of Equation (20) can be obtained:

$$\overset{\bullet}{V} = S\overset{\bullet}{S} = \begin{bmatrix} \widetilde{i_\alpha} & \widetilde{i_\beta} \end{bmatrix} \begin{bmatrix} \frac{d\widetilde{i_\alpha}}{dt} \\ \frac{d\widetilde{i_\beta}}{dt} \end{bmatrix} \leq 0 \tag{22}$$

Substituting Equation (19) into Equation (22), the following equations can be obtained:

$$\begin{cases} \widetilde{i}_\alpha * \frac{d\widetilde{i_\alpha}}{dt} + \widetilde{i}_\beta * \frac{d\widetilde{i_\beta}}{dt} = -\frac{R}{L}\left[(\widetilde{i_\alpha})^2 + (\widetilde{i_\beta})^2\right] & K\left|\widetilde{i_s}\right| > a_0 \\ +\frac{1}{L}\widetilde{i}_\alpha[E_\alpha - Ksign(\widetilde{i}_\alpha) + \frac{1}{L}\widetilde{i}_\beta(E_\beta - Ksign(\widetilde{i}_\beta))] & \\ \widetilde{i}_\alpha * \frac{d\widetilde{i_\alpha}}{dt} + \widetilde{i}_\beta * \frac{d\widetilde{i_\beta}}{dt} = -\frac{R}{L}\left[(\widetilde{i_\alpha})^2 + (\widetilde{i_\beta})^2\right] & K\left|\widetilde{i_s}\right| \leq a_0 \\ +\frac{1}{L}\widetilde{i}_\alpha[E_\alpha - K\left|\widetilde{i}_\alpha\right|\arcsin(\frac{\lambda}{\varepsilon}(\widetilde{i}_\alpha)) + \frac{1}{L}\widetilde{i}_\beta(E_\beta - K\left|\widetilde{i}_\alpha\right|\arcsin(\frac{\lambda}{\varepsilon}(\widetilde{i}_\alpha))] & \end{cases} \tag{23}$$

The simplified formula of Equation (22) is as follows:

$$\begin{cases} \widetilde{i}_\alpha * \frac{d\widetilde{i_\alpha}}{dt} + \widetilde{i}_\beta * \frac{d\widetilde{i_\beta}}{dt} = \left|\widetilde{i}_\alpha\right|[-\frac{R_s}{L_s}\left|\widetilde{i}_\alpha\right| + \frac{1}{L_s}E_\alpha - \frac{1}{L_s}Ksign(\widetilde{i}_\alpha)] & K\left|\widetilde{i_s}\right| > a_0 \\ +\left|\widetilde{i}_\beta\right|[-\frac{R_s}{L_s}\left|\widetilde{i}_\beta\right| + \frac{1}{L_s}E_\beta - \frac{1}{L_s}Ksign(\widetilde{i}_\beta)] & \\ \widetilde{i}_\alpha * \frac{d\widetilde{i_\alpha}}{dt} + \widetilde{i}_\beta * \frac{d\widetilde{i_\beta}}{dt} = \left|\widetilde{i}_\alpha\right|[-\frac{R_s}{L_s}\left|\widetilde{i}_\alpha\right| + \frac{1}{L_s}E_\alpha - \frac{1}{L_s}K\left|\widetilde{i}_\alpha\right|\arcsin(\frac{\lambda}{\varepsilon}(\widetilde{i}_\alpha))] & K\left|\widetilde{i_s}\right| \leq a_0 \\ +\left|\widetilde{i}_\beta\right|[-\frac{R_s}{L_s}\left|\widetilde{i}_\beta\right| + \frac{1}{L_s}E_\beta - \frac{1}{L_s}K\left|\widetilde{i}_\beta\right|\arcsin(\frac{\lambda}{\varepsilon}(\widetilde{i}_\beta))] & \end{cases} \tag{24}$$

When Equation (24) is less than 0, the system enters a steady state, that is, when satisfying the following formula, the system is stable.

$$\begin{cases} K > \dfrac{-R\left|\widetilde{i_s}\right| + E_s}{sign(\widetilde{i_s})} & K\left|\widetilde{i_s}\right| > a_0 \\ \\ K > \dfrac{-R\left|\widetilde{i_s}\right| + E_s}{\left|\widetilde{i}_s\right|\arcsin(\frac{\lambda}{\varepsilon}(\widetilde{i}_s))} & K\left|\widetilde{i_s}\right| \leq a_0 \end{cases} \tag{25}$$

According to the sliding mode reaching condition $\widetilde{i} \bullet \overset{\bullet}{\widetilde{i}} < 0$, the value range of gain $K$ can be calculated as follows:

$$\begin{cases} K > \max\left\{\dfrac{-R\left|\widetilde{i_\alpha}\right| + E_\alpha}{sign(\widetilde{i_\alpha})}, \dfrac{-R\left|\widetilde{i_\beta}\right| + E_\beta}{sign(\widetilde{i_\beta})}\right\} & K\left|\widetilde{i_s}\right| > a_0 \\ \\ K > \max\left\{\dfrac{-R\left|\widetilde{i_\alpha}\right| + E_\alpha}{\left|\widetilde{i}_\alpha\right|\arcsin(\frac{\lambda}{\varepsilon}(\widetilde{i}_\alpha))}, \dfrac{-R\left|\widetilde{i_\beta}\right| + E_\beta}{\left|\widetilde{i}_\beta\right|\arcsin(\frac{\lambda}{\varepsilon}(\widetilde{i}_\beta))}\right\} & K\left|\widetilde{i_s}\right| \leq a_0 \end{cases} \tag{26}$$

Therefore, when *K* satisfies Equation (26), the new observer system will converge rapidly in the global range.

## 5. Simulation and Experimental Verification

### 5.1. Simulation Model and Parameters

The simulation model of the sensorless vector control of a PMSM is established. In order to ensure the fairness of the comparison, the bus voltage is set to 311 V, and the PWM output frequency to the rated working frequency of the switch. The simulation parameters are shown in Table 1. Under the same simulation parameters, the performances of the traditional SMO and the new SMO system are compared to verify the superiority of the new SMO. The system control block diagram is shown in Figure 3 below, and the motor parameters are shown in Table 2. The PI parameters of both current loops are the same as: $k_p = 28, k_i = 9583$. The PI parameters of the speed loop are $k_p = 1, k_i = 1.5$. The gain of the observer is $k = 200$. The results of the simulation are shown in Figures 4–9.

**Table 1.** Simulation parameters.

| Parameters | Values |
|---|---|
| PWM switching frequency $f_{PWM}$ | 10 kHZ |
| Sample time (s) | $2*10^{-7}$ |
| Starting rotor speed (r/min) | 0 |
| Reference rotor speed (r/min) | 1000 |
| Starting load torque (N·m) | 0 |
| Reference load torque (N·m) | 10 |

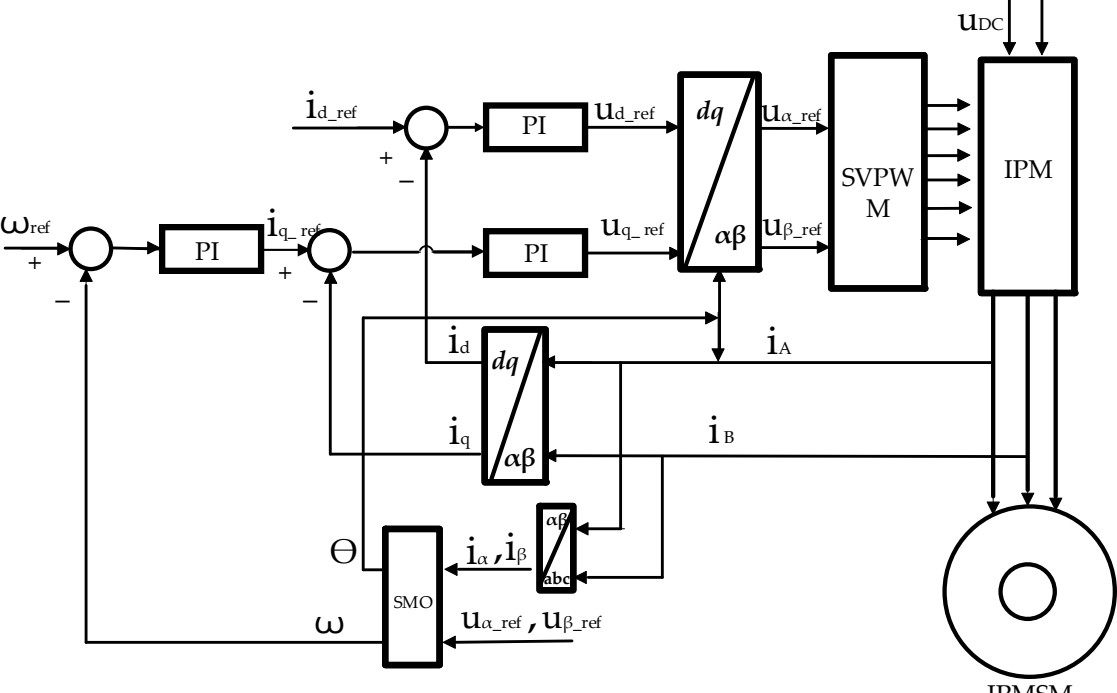

**Figure 3.** Structure diagram of the sensorless vector control system of permanent magnet synchronous motor.

**Table 2.** PMSM parameters.

| Parameters | Values |
| :---: | :---: |
| Pole pairs | 4 |
| Stator inductance $L_s/H$ | 0.0085 |
| Stator resistance $R_s/\Omega$ | 2.8750 |
| Flux linkage $\varphi_f/Wb$ | 0.175 |
| Rotational inertia J/kg·m$^2$ | 0.008 |
| Damping | 0.0003 |

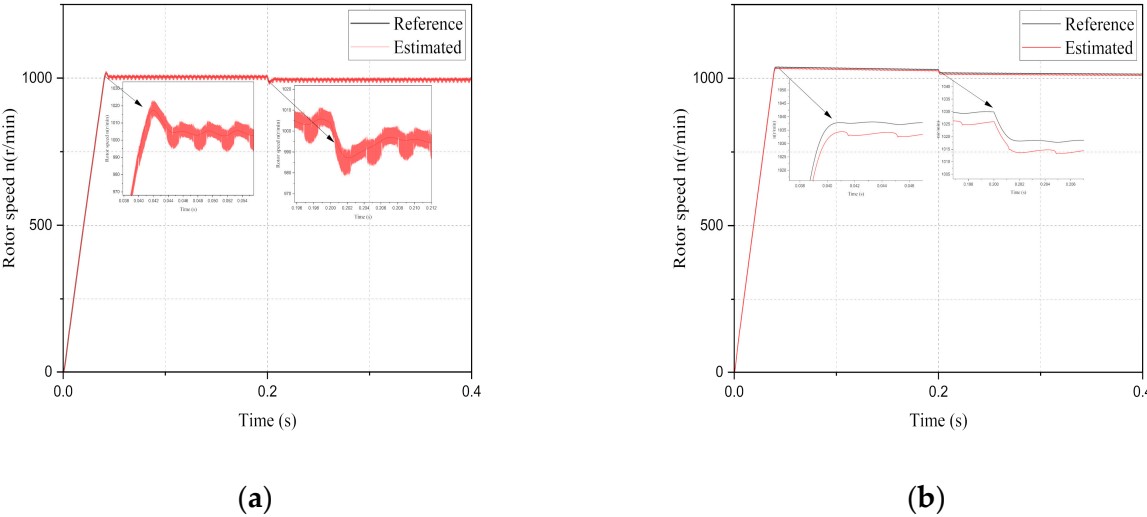

(**a**)                                    (**b**)

**Figure 4.** Simulation results of rotor speed. (**a**) Traditional SMO, (**b**) new SMO.

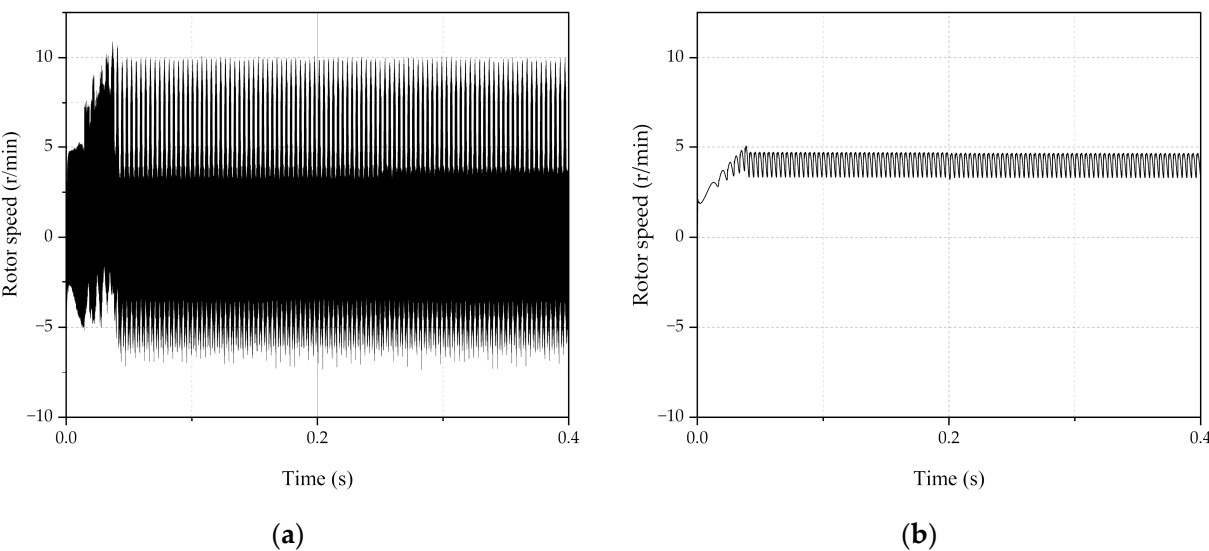

(**a**)                                    (**b**)

**Figure 5.** Simulation results of speed error. (**a**) Traditional SMO, (**b**) new SMO.

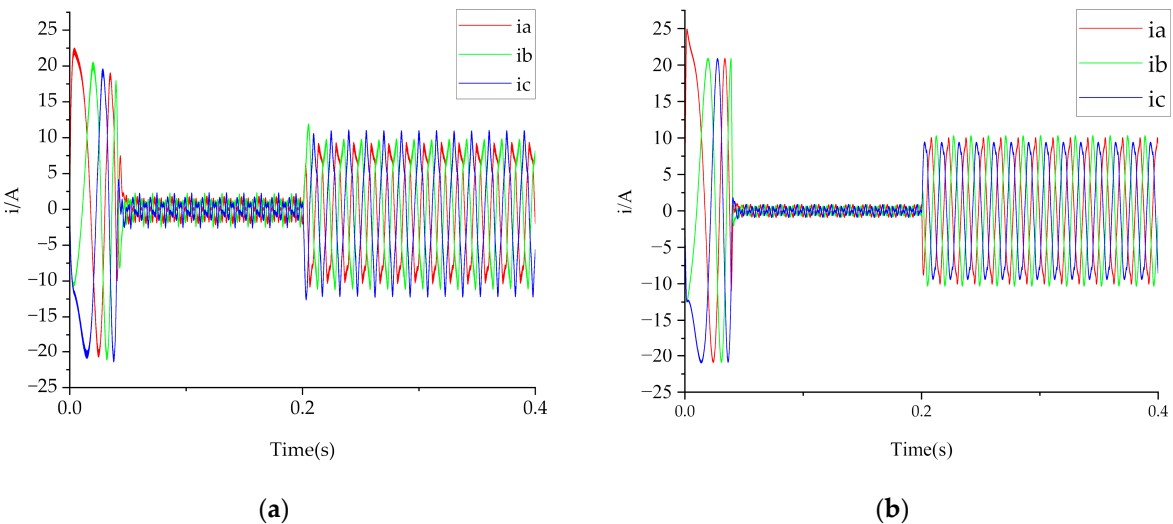

**Figure 6.** Simulation results of three-phase current. (**a**) Traditional sliding mode observer, (**b**) new sliding mode observer.

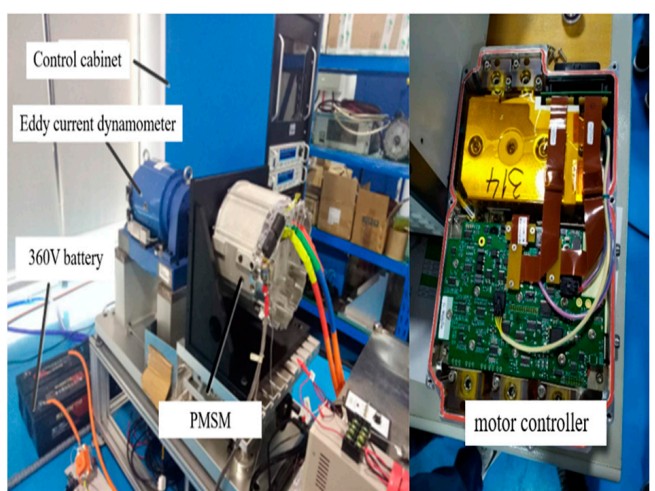

**Figure 7.** Experimental platform and motor controller.

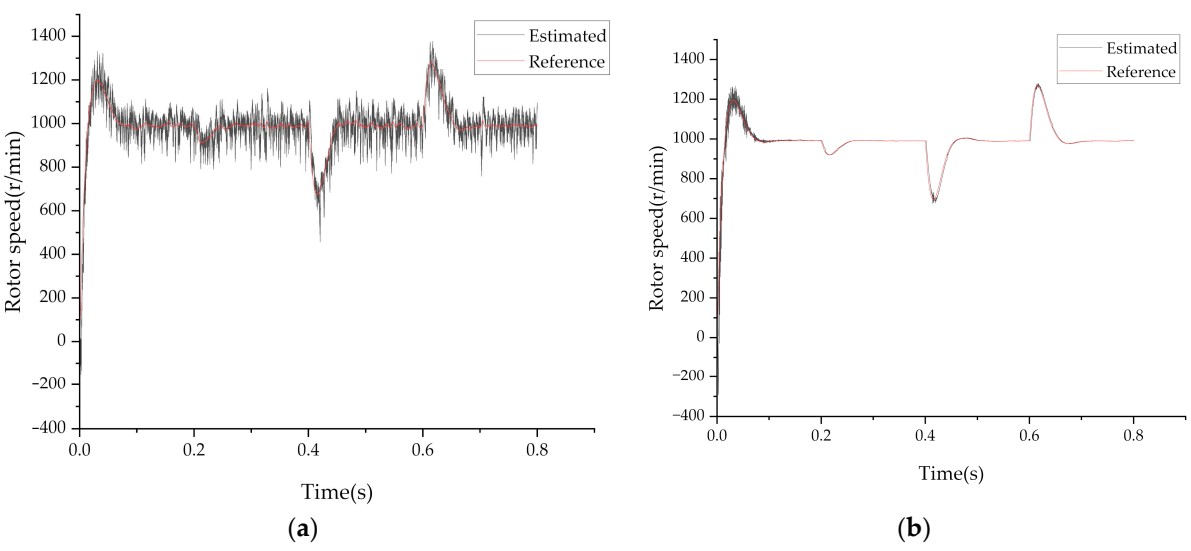

**Figure 8.** Experimental results of rotor speed. (**a**) Traditional SMO, (**b**) new SMO.

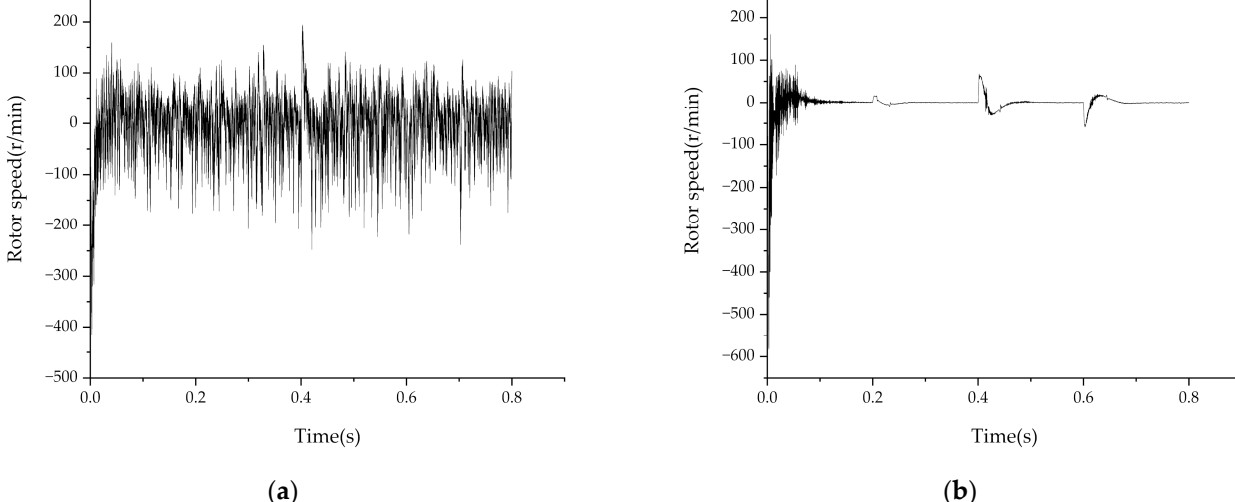

**Figure 9.** Experimental results of rotor speed error. (**a**) Traditional sliding mode observer, (**b**) new sliding mode observer.

*5.2. Analysis of Simulation Results*

The rotor speed simulation waveforms are shown in Figure 4. In the process of speed observation, when the speed of the motor rises from 0 to the given speed 1000 r/min, the adjustment time of the traditional sliding mode observer is $T = 0.05$ s and the overshoot is $\sigma = 3\%$. After reaching the given speed, the chattering phenomenon is large. When running to 0.2 s, the load torque is suddenly increased to 10 N·m, and the speed overshoot is 1%. Compared with the above indicators, the adjustment time of the new sliding mode observer is $T = 0.04$ s, and the overshoot is 4%. After reaching the given speed, the chattering phenomenon is very small. When running to 0.2 s, the load torque is suddenly increased to 10 N·m, and the speed overshoot is $\sigma = 0.2\%$. Thus, the effectiveness of the combined reaching law is verified. The comparative simulation results of the startup can be seen in Table 3.

**Table 3.** The comparative simulation results of startup.

| Method | Reference Speed | Time to Reach Steady State | Overshoot |
|---|---|---|---|
| Traditional SMO | 1000/min | 0.05 s | $\sigma = 3\%$ |
| New SMO | 1000/min | 0.04 s | $\sigma = 0.2\%$ |

Figure 5 shows the simulation plots of the estimated speed and the actual speed error of the SMO based on the sign function and the SMO based on the combined convergence law, respectively, from which it can be seen that when the motor is just started, the speed error observed by the SMO based on the sign function fluctuates sharply –7.5 and 10 r/min and the jitter vibration phenomenon is very serious, and when $T = 0.2$ s is given as the load, the load change is not sensitive to the speed error changes. The sliding mode observer based on the combined convergence law observes that the speed error fluctuates within a range of 3 ∼ 5 r/min, with a significant reduction in jitter, and when running to 0.2 s, the load torque is suddenly increased to 10 N·m and leads to a slight reduction in speed error, thus verifying that the combined convergence law SMO has good robustness. A comparison of the simulation results of rotational speed error can be seen in Table 4.

**Table 4.** Comparison of simulation results of rotational speed error.

| Method | Rotational Speed Error during Motor Start |
|---|---|
| Traditional SMO | $-7.5 \sim 10 \, \text{r/min}$ |
| New SMO | $3 \sim 5 \, \text{r/min}$ |

From the three-phase current simulation curves in Figure 6, it can be seen that the three-phase currents estimated by the traditional sliding mode fluctuate between –20 and 20 A at the beginning and tend to stabilize after 0.05 s with a fluctuation between –2 and 2 A. After the load torque is added, the three-phase currents present basic sine waves with large harmonics. The three-phase currents estimated by the new SMO fluctuate in the range of $-20 \sim 20$ A at the beginning and tend to be stable after 0.04 s with a fluctuation in the range of $-1 \sim 1$ A. After the load torque is added, the three-phase currents present standard sine waves with small harmonics. The comparative simulation results of the startup can be seen in Table 5.

**Table 5.** Comparison of simulation results of three-phase current.

| Method | Starting Current of Motor | Time to Reach Steady State | Current Range Under Sudden Load |
|---|---|---|---|
| Traditional SMO | $-20 \sim 20$ A | 0.05 s | $-2 \sim 2$ A |
| New SMO | $-20 \sim 20$ A | 0.04 s | $-1 \sim 1$ A |

*5.3. Experimental Validation*

As illustrated in Figure 7, the new SMO is applied to the experimental platform to test its effectiveness. The motor is a 20 KW PMSM, and the switching frequency of the motor controller is 10 kHz. The boundary layer thickness of the combined reaching law is taken as $\varepsilon = 1.5$, and the switching gain of $k$ is taken as 200 based on the stability circumstances and relevant motor parameters.

Under the identical experimental conditions, the performance of a traditional observer's control system and the new observer's control system are compared. The given reference speed is 1000 r/min. By applying a minor load of 2 N · m at 0.2 s, raising the load to 10 N · m at 0.4 s, and reducing the load from 10 N · m to 2 N · m at 0.6 s, the experimental effects of the control system are noticed.

Figure 8 demonstrates the experimental findings of a traditional SMO based on the sign switching function, as well as a PMSM vector control system based on the new SMO. Compared to the traditional SMO, the chattering phenomena are greatly lessened, the speed estimation is more precise, and it also has a better speed estimation capacity while coping with unexpected load changes after adopting the new sliding mode reaching law. The comparative experimental results of the startup can be seen in Table 6.

**Table 6.** The comparative experimental results of startup.

| Method | Reference Speed | Time to Reach Steady State | Overshoot |
|---|---|---|---|
| Traditional SMO | 1000/min | 0.1 s | $\sigma = 8\%$ |
| New SMO | 1000/min | 0.08 s | $\sigma = 0.5\%$ |

The speed errors between the observed and actual values change dramatically in the range of $-150 \sim 150$ r/min based on the typical sliding mode observer, as shown in Figure 9, particularly when the load increases and lowers quickly. Based on the new sliding mode observer, the speed errors vary greatly at first, and when the load increases and drops suddenly, the speed errors vary in the range of $-60 \sim 60$ r/min, but after the system enters the steady state, the speed errors quickly tend to 0. The observed speed errors using

the new sliding mode observer are decreased by 80% compared to the standard sliding mode observer, and the chattering problem is considerably improved. A comparison of the experimental results of rotational speed error can be seen in Table 7.

**Table 7.** Comparison of experimental results of rotational speed error.

| Method | Rotational Speed Error during Motor Start |
|---|---|
| Traditional SMO | $-150 \sim 150\,\mathrm{r/min}$ |
| New SMO | $-60 \sim 60\,\mathrm{r/min}$ |

It can be seen from Figure 10 that the three-phase currents based on the traditional sliding mode observer control system can present the basic sinusoidal current changes, but the currents have a certain distortion and are more inclined to the triangular wave, and the maximum overshoot of the currents reaches 40 A. The three-phase current changes based on the new sliding mode observer present close to the regular sine waves with relatively low harmonics, and the maximum overshoot of the currents is 32 A. The comparative experimental results of the startup can be seen in Table 8.

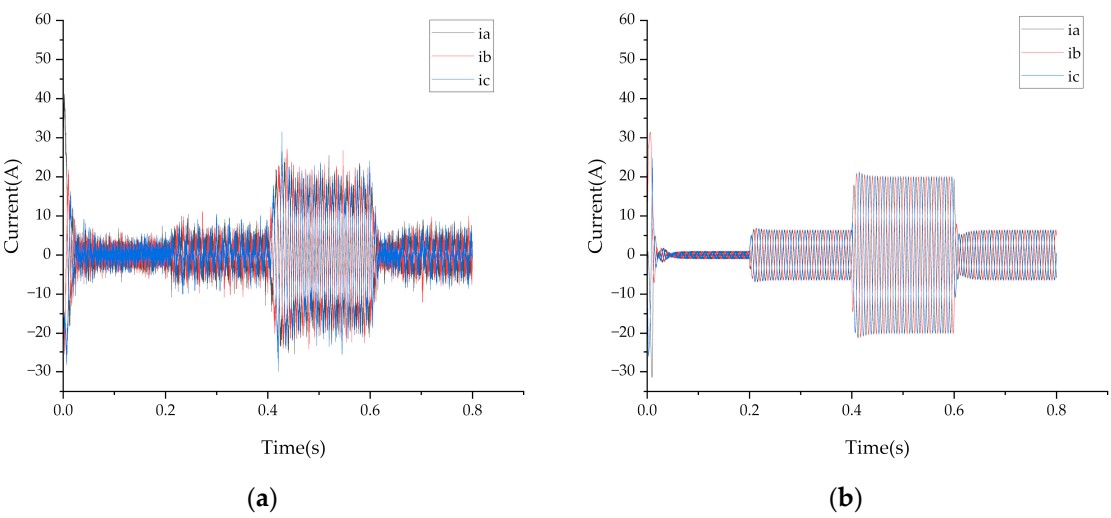

**(a)**            **(b)**

**Figure 10.** Experimental results of three-phase current. (**a**) Traditional sliding mode observer, (**b**) new sliding mode observer.

**Table 8.** Comparison of experimental results of three-phase current.

| Method | Starting Current of Motor | Time to Reach Steady State | Current Range Under Sudden Load |
|---|---|---|---|
| Traditional SMO | $-30 \sim 40\,\mathrm{A}$ | 0.05 s | $-10 \sim 10\,\mathrm{A}$ |
| New SMO | $-30 \sim 30\,\mathrm{A}$ | 0.04 s | $-1 \sim 1\,\mathrm{A}$ |

## 6. Conclusions and Future Works

To mitigate the jitter problem in sliding mode control, this study proposes a combined convergence law algorithm based on the exponential convergence law and variable speed convergence law to suppress the system's jitter problem. The comparative experimental results reveal that the suggested novel combined convergence law efficiently improves observation accuracy while drastically weakening the system's jitter problem.

In this paper, although some of the above research results have been obtained, further in-depth studies can still be carried out from this aspect: no specific analysis process is given on the selection of switching points in the combined reaching law.

**Author Contributions:** Conceptualization, data collection, and representation, review—Z.L. writing—original draft, editing, and data collection—W.C. All authors have read and agreed to the published version of the manuscript.

**Funding:** This work was supported by the National Science Foundation of China (Project 11572055). And the Scientific Research Fund of Hunan Provincial Education Department (Project No. 18C0219), Qingqing Xiang for funding support.

**Informed Consent Statement:** Not applicable for this work.

**Data Availability Statement:** Not applicable for this work.

**Conflicts of Interest:** The authors do not have any conflict of interest.

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
