# Peer review of "Research on an Improved Sliding Mode Observer for Speed Estimation in Permanent Magnet Synchronous Motor"

_processes, doi:10.3390/pr10061182_

Round 1

Reviewer 1 Report

The work is well written. The idea expressed in the conclusions is very good and they can continue the research in this regard. 

Author Response

Response to Editor and Reviewers Comments

Dear Editor(s) of Processes,

Title: Research on an Improved Sliding Mode Observer for Speed Estimation in Permanent Magnet Synchronous Motor

Reviewer 2 Report

The manuscript presents a novel strategy for the Sliding Mode Observer to control the permanent magnet synchronous motor. This strategy is composed of two different functions, an exponential law and an arcsine function. The experimental results show a speed error reduction compared with the traditional sliding mode observer.

In general, the results are promising; however, the presentation is what needs improvement. There are some typos in equations due to indexes that make them difficult to follow (below more details). Graphs show promising results, however, some are difficult to visualise even when zooming in.

Line 44 - Acronym IPMSM is not defined.

Line 46 - Change the "," by a ".".

Line 50-51 Substitute "permanent magnet synchronous motor" by the acronym PMSM.

Equation 1 - There may be a mistake with the subindices for "I".

Equation 2 is incorrect due to the mistake on subindices on equation 1.

Line 58 -  re->are

Remove the braces on equations 1, 7, 8, and 10. The "cases" environment should be used only to define a function by "parts", as done in equation 15.

Line 71 - remove the word "Symbol".

Line 83 - check the subindex for i_s.

Equations 9 and 16 - What is S^{\dot}. Are you referring to the derivative? If that is the case, then use the same notation as in the previous equations, do not mix Newton's and Leibniz's notation.

Line 142 - Change "," by ".".

Line 186 - Check referred equations.

Line 221 - "5 simulation" --> "5 Simluation".

Line 223 and 224 - Use the acronym PMSM.

Line 226 - There is no need to indicate a zero multiplication. You may substitute by 0.

Parameter values on lines 223 - 234 should be presented as a separate table.

Section 5.2 should be improved. A set of Tables and better figures may enhance the presentation of the results.

Line 261 - "and When" --> "and when".

The future work section may be improved, and it should be mentioned "why" the proposed work was not carried out. The current title of the manuscript, "Research on an Improved Sliding Mode Observer for PMSM", is so general that the proposed future work should be included as part of the research. I recommend changing the title to something more specific that clearly states the boundaries of the manuscript.

Author Response

Response to Editor and Reviewers Comments

Dear Editor(s) of Processes,

Title: Research on an Improved Sliding Mode Observer for Speed Estimation in Permanent Magnet Synchronous Motor

Author:

 I am sending a copy of the revised manuscript of the above referenced paper for your consideration for publication in Processes. The paper has been carefully amended incorporating and addressing the comments made by editor and reviewers as appended below.
